# Structural Characterization of Pinnatoxin Isomers

**DOI:** 10.3390/md23030103

**Published:** 2025-02-26

**Authors:** Andrew I. Selwood, Christopher O. Miles, Alistair L. Wilkins, Frode Rise, Sarah C. Finch, Roel van Ginkel

**Affiliations:** 1Cawthron Institute, Nelson 7010, New Zealand; roel.vanginkel@cawthron.org.nz; 2Norwegian Veterinary Institute, Elizabeth Stephansens vei 1, 1433 Ås, Norway; christopher.miles@vetinst.no (C.O.M.); wilkinsalw@hotmail.com (A.L.W.); 3Chemistry Department, The University of Waikato, Hamilton 3240, New Zealand; 4Department of Chemistry, University of Oslo, Blindern, P.O. Box 1033, NO-0315 Oslo, Norway; frode.rise@kjemi.uio.no; 5AgResearch Ltd., Ruakura Research Centre, Hamilton 3240, New Zealand; sarah.finch@agresearch.co.nz

**Keywords:** pinnatoxin, pteriatoxin, cyclic imine, *Vulcanodinium rugosum*, NMR, LC–MS, LD_50_

## Abstract

Pinnatoxins, a group of marine biotoxins primarily produced by the dinoflagellate *Vulcanodinium rugosum*, have garnered significant attention due to their potent toxic effects and widespread distribution in marine ecosystems. LC–MS analysis of shellfish and *V. rugosum* cultures revealed the presence of previously unidentified isomers of pinnatoxins D, E, F, and H, at levels approximately six times lower than those of known isomers. The chemical structures of these isopinnatoxins were determined using a combination of LC–MS/MS and NMR spectroscopy, which demonstrated that the isomerization of each pinnatoxin occurred through the opening and recyclization of the spiro-linked tetrahydropyranyl D-ring to form a smaller tetrahydrofuranyl ring. The acute toxicity of isopinnatoxin E was determined by intraperitoneal injection into mice and was found to be significantly lower than that of pinnatoxin E. Given their low toxicity and low abundance, it is unlikely that isopinnatoxins contribute significantly to the overall toxicity of pinnatoxins.

## 1. Introduction

Pinnatoxins (Figure 1), are a group of cyclic imines that have distinctive structural features, including a macrocycle incorporating a dispiroketal (BCD rings), a bicycloketal (EF rings), and a cyclic imine (A ring) that is spirolinked to a cyclohexene (G ring) [1,2,3,4,5]. Pinnatoxins E (**1**), F (**2**), G (**9**), and H (**4**) are produced by the benthic dinoflagellate *Vulcanodium rugosum*, and strain-dependent toxin profiles have been found in numerous isolates from around the world [6,7,8,9,10,11,12]. These toxins can accumulate in a variety of bivalves, and pinnatoxins A–D, along with pteriatoxins A–C [2], and pinnatoxin fatty acid esters [13], are thought to be shellfish metabolites of pinnatoxins produced by *V. rugosum*.

Pinnatoxins are highly toxic to mice by intraperitoneal (i.p) injection and oral administration [14]. They are fast-acting neurotoxins that target muscle and neuronal nicotinic acetylcholine receptors [15], leading to death by respiratory failure at high doses. Despite the frequent detection of these potent neurotoxins in shellfish, no human intoxications have been reported [16,17,18]. Ongoing investigations into the occurrence, chemistry, and toxicology of these toxins continue to refine our understanding of the potential risks they pose to shellfish consumers.

In this study, we expanded on the previous discovery of unidentified pinnatoxin isomers in pacific oysters, razor fish, sediment and water samples [2]. These pinnatoxin isomers were previously reported as *epi*-pinnatoxin E and *epi*-pinnatoxin F, but are now referred to as isopinnatoxin E (**5**) and isopinnatoxin F (**6**). Here, we report on the occurrence of isopinnatoxins in cultures of *V. rugosum*, the acid-catalyzed isomerization of pinnatoxins, and the isolation and structural elucidation of isopinnatoxin E (**5**). Furthermore, the acute toxicity of **5** was determined in mice by i.p. injection.

## 2. Results

### 2.1. Structural Charaterization of Isopinnatoxin E

Isopinnatoxin E (**5**) (290 µg) was obtained as a colorless, amorphous solid. The collision-induced fragmentation mass spectrum of **5** showed no significant differences from that of **1** (Figure 2), suggesting that only a minor structural rearrangement had occurred.

The structure of isopinnatoxin E (**5**) was established by the examination of 1- and 2-D NMR spectra (Appendix A), together with a series of isomerization experiments. The ^13^C APT and g-HSQC spectra of **5** revealed the presence of 5 methyl (4 doublets and one singlet), 18 methylene, 13 methine, and 6 quaternary carbon atoms. Three further quaternary carbon signals were observed in the g-HMBC spectrum at 179.9 ppm (C-6), 148.3 ppm (C-10), and 182.2 ppm (C-39). Correlations observed in the two-dimensional NMR spectra established the assignments for **5** (Table 1), in a manner analogous to that used to establish the identity and NMR assignments of **1** [2]. The majority of the resonances of **5** possessed very similar chemical shifts and multiplicities to those of the analogous resonances of **1**. However, the ^1^H and ^13^C resonances associated with C-19–C-24, and the ^1^H resonances associated with C-1–C-4, C-7–C-9, C-11, and C-12, differed significantly from the equivalent resonances of **1**. The chemical shifts of many of these resonances more closely resembled those of spirolide C (**10**) (Table 1) than they did **1** (Appendix A).

Of particular importance in determining the structure of **5** were the correlations observed in the HMBC and NOESY spectra (Figure 3 and Appendix A). Although H-23 of **5** showed HMBC correlations to C-21, C-24, and C-25, it did not show a correlation to C-19 as might be expected if the tetrahydropyranyl D-ring had remained intact. Furthermore, H-22 of **5** showed HMBC correlations to C-19, C-21, and 21-CH_3_, establishing the presence of a tetrahydrofuranyl ring with an ether linkage connecting C-19 and C-22 to form a dispiro[4.1.5.2] spiroketal system. The strong NOESY correlation observed between H-21 and H-23 of **5** confirms that they are on the same face of the molecule, opposite to H-22, as do the observed coupling constants for H-23 (Table 1, Appendix A). This indicates 23*S*-stereochemistry, consistent with that of compound **1**. These and other correlations established the structure of **5** to be that shown in Figure 1.

### 2.2. Isomerization

Upon heating in aqueous acetonitrile (MeCN) solutions containing trifluoroacetic acid (TFA), the D-ring in compounds **1**–**5** rapidly rearranged to form equilibrated mixtures of pinnatoxin isomers with the thermodynamically favored tetrahydropyranyl form being the major isomer (85:15) (Table 2, Figure 4 and Appendix A). However, this isomerization did not occur in compound **9**, attributable to the absence of the 22-OH group. The equilibrated ratio of the isomers under these conditions was consistent with the ratio observed in extracts of *V. rugosum* (Figure 5).

### 2.3. Toxicity of Isopinnatoxin E

The acute toxicity of **5** by i.p. injection to mice was determined according to the principles of OECD guideline 425 [20]. Mice dosed with 280 (1 mouse), 400 (1), 700 (1) or 980 (3) µg/kg all survived. Observed symptoms of toxicity included abdominal stretching and orbital tightening, as well as a flat posture with the ears back. These symptoms were evident within 10 min post-dosing and were resolved within 2 h. During the subsequent 14-day observation period, all surviving mice exhibited normal growth and feeding behavior. At necropsy no abnormalities were detected, and organ weights were within normal limits. Mice (2) dosed at 1390 µg/kg both died with death times of 12- and 7-min post-dosing. The median lethal dose (LD_50_) for **5** by i.p. injection was calculated to be 1250 µg/kg with a 95% confidence interval of 980 to 1390 µg/kg.

## 3. Discussion

The initial discovery of low levels of compounds **5** and **6** in environmental samples, along with the observation that they were formed in acidic solutions [2], prompted an investigation to determine the chemical structures of these compounds. We hypothesized that the hydroxy group at C-22 was involved in the isomerization because the isomerization only occurred in pinnatoxins which contained a hydroxy group at that position. Cultures of *V. rugosum* were found to contain isopinnatoxins, which enabled the isolation and characterization of **5**. The structure of **5** was elucidated from 1-D and 2-D NMR spectra, confirming that a rearrangement of the D-ring spiroketal with the 22-OH had occurred. The equilibrium favored the six-membered tetrahyropyranyl D-ring in pinnatoxins over the five-membered tetrahydrofuranyl D-ring in isopinnatoxins, with isopinnatoxins comprising only 15% of the total.

In addition to characterizing the structures of isopinnatoxins, experiments were conducted to provide insight into how they might contribute to the toxicity of shellfish containing pinnatoxins. The LD_50_ of **5** (1250 µg/kg) was significantly lower than that of 1 (57 µg/kg) [14]. This result parallels the diminished toxicity observed for another class of dinoflagellate toxins, pectenotoxins, where the rearrangement of a spiroketal ring system greatly reduces toxicity [21]. Both pinnatoxins and pectenotoxins consist of polyether rings within a macrocycle. Changes to the spiro-ether ring structures alter the overall conformation of the macrocycles, which likely explains the reduction in toxicity. Based on this structure–activity relationship, it is likely that the other isopinnatoxins will also exhibit a similar reduction in toxicity.

Two different health-based guidance values have been proposed for pinnatoxins in shellfish: 23 μg/kg by Arnich et al. [17] and 268 μg/kg by Finch et al. [16], with the discrepancy due to the application of different uncertainty factors. The i.p toxicities of **1**, **2**, **9,** and **4** were, respectively, 27, 98, 25, and 19 times greater than that of **5**, respectively. Given the low toxicity and low abundance of isopinnatoxins, it is unlikely that they could make any meaningful contribution toward the health guidance figures set for total pinnatoxins.

## 4. Materials and Methods

### 4.1. Purification of Isopinnatoxin E

Cells from a culture of *V. rugosum* isolate CAWD167 were lyophilized and extracted with methanol (MeOH) containing 0.1% (*v*/*v*) acetic acid. The extract was evaporated to dryness in vacuo, and the resulting residue was dissolved in ethanol–H_2_O (1:1 *v*/*v*) and defatted with an equal volume of hexane. The ethanolic layer was then extracted with an equal volume of ethyl acetate (EtOAc), and the upper organic layer was evaporated to dryness in vacuo. The residue was then dissolved in 10% MeOH–H_2_O (1:9 *v*/*v*) containing 0.1% (*v*/*v*) acetic acid and dichloromethane (DCM). The DCM was separated and evaporated in vacuo, and the residue was dissolved in MeCN–H_2_O (1:4 *v*/*v*) in aqueous 100 mM triethylamine.

This solution was left for 24 h at 21 °C to hydrolyze compounds **2** and **6** to compounds **1** and **5**. Water was then added to the solution to lower the MeCN concentration to approximately 10% (*v*/*v*), and the pH was adjusted to pH 5.5 using acetic acid. The solution was desalted by applying it to a 200 mg Strata-X solid phase extraction (SPE) column (Phenomenex, Torrance, CA, USA), pre-conditioned with 6 mL MeOH, followed by 6 mL MeCN–H_2_O (3:17 *v*/*v*). The column was washed with 6 mL MeCN–H_2_O (3:17 *v*/*v*), and compound **5** was eluted with 5 mL MeOH.

The MeOH was evaporated under a gentle stream of nitrogen, and the residue was dissolved in 2 mL DCM. This was loaded onto a 5 g silica gel column (32–63 µm, Scientific Adsorbents Inc., Atlanta, GA, USA), which had been pre-conditioned with 20 mL DCM. The column was eluted using a stepwise gradient of MeOH in DCM (5%, 10%, 20%, 25%, 30%, 40%, 60%, 80% and 100%; 4 × 10 mL each). Fractions 12–30 were combined and evaporated in vacuo.

The residue was then dissolved in 5 mL MeOH–H_2_O (1:3 *v*/*v*) and applied to a 200 mg Strata-X SPE column pre-conditioned with 6 mL MeOH, followed by 6 mL MeOH–H_2_O (1:3 *v*/*v*). The SPE column was eluted using a stepwise gradient of MeCN in aqueous 10 mM NaH_2_PO_4_ (20%, 25%, 30% and 40%; 3 × 7.5 mL each). Fractions 10–12 were combined and partially evaporated in vacuo to reduce the MeCN content prior to desalting on a 200 mg Strata-X SPE column using the previously described protocol.

Final purification was achieved by preparative HPLC using a Luna C18(2) 10 µm column (250 × 10 mm, Phenomenex Torrance, CA, USA) eluted at 5 mL/min with 40% MeCN in aqueous 10 mM NaH_2_PO_4_, and was monitored using a photodiode array detector scanning 200–300 nm. Compound **5** was collected from 10.1 to 11.2 min. The solution of **5** was partially evaporated to reduce the MeCN content prior to desalting on a 200 mg Strata-X SPE column, following the previously described protocol.

### 4.2. LC–MS Analysis

A Shimadzu LCMS-8050 with Nexera Series LC-30 UHPLC (Shimadzu, Kyoto, Japan) was used for analysis. Separation was achieved with a Waters Acquity BEH C18 1.7 µm 50 × 2.1 mm column (Phenomenex, Torrance, CA, USA) at 30 °C, eluted at 0.5 mL/min with a linear gradient of 0.1% (*v*/*v*) formic acid in water (A) to MeCN (B). The gradient consisted of 20 to 60% B over 2 min, held for 8 min before returning to the initial conditions. The electrospray ionization was operated in positive ion mode with the following settings: interface voltage 4 kV, nebulizing gas flow 2 L/min, heating and drying gas flows 10 L/min, interface temperature 300 °C, desolvation temperature 525 °C, DL temperature 250 °C, heat block temperature 400 °C, and CID gas 270 kPa. For MS/MS experiments, the collision energy was set to −50 eV. The following single-ion monitoring (SIM) transitions were acquired with 20 ms dwell times: *m/z* 694.5, 708.5, 766.5, 782.5, and 784.5. For the analysis of *V. rugosum* cultures, multiple-reaction monitoring (MRM) transitions were acquired with 20 ms dwell times: *m/z* 694.5→164.1, 708.5→164.1, 766.5→164.1, 784.5→164.1. The *m*/*z* 164.1 ion corresponds to the common iminium ring fragment (C_11_H_18_N^+^). 

The concentration of **5** was determined by calibrating against the certified reference material of **1** (Sigma–Aldrich, Merck Life Science Ltd., Buchs, Switzerland) using the same SIM LC–MS method, using an isocratic elution consisting of 35% mobile phase B. An equimolar response of **1** and **5** under these conditions was confirmed by the quantitative isomerization experiment.

### 4.3. Pinnatoxin Isomerization

Solutions of **1**–**5** and **9** were prepared in MeCN–water–TFA (1:1:0.2). The solutions were heated to 50 °C in a chemical reaction oven, and samples were taken at 0, 5, 20, and 60 min for LC–MS/MS analysis. These samples were diluted 100-fold by adding 10 µL aliquots to 990 µL of MeCN–water (1:1 *v*/*v*) and analyzing immediately by LC-MS/MS.

### 4.4. NMR Spectroscopy

NMR (^1^H, COSY, TOCSY, g-HSQC, g-HMBC, NOESY, and ^13^C APT) spectra were obtained from solutions of CD_3_OD (99.8+ atom-% D; Aldrich) using a Bruker Avance AVI-600 MHz NMR spectrometer, with a 5 mm inverse TCI (^1^H^13^C^15^N) cryoprobe, equipped with Z-gradient coils, at 30 °C, using TopSpin version 1.3. Chemical shifts are reported relative to internal C*H*D_2_OD (3.31 ppm) and *C*D_3_OD (49.0 ppm) and NMR data were processed using TopSpin version 3.6.

### 4.5. Toxicity Determination for Isopinnatoxin E

Acute toxicity was determined according to the principles of OECD guideline 425 [20]. This guideline uses the minimum number of animals while still being able to determine a robust LD_50_, along with an estimate of confidence intervals. Mice (female, Swiss, 18–22 g) were weighed immediately prior to dosing, and **5** was administered on a µg/kg basis. Aliquots of an EtOH solution containing **5** were taken gravimetrically and diluted with 1% Tween in saline prior to dosing (500 µL). All mice were dosed between 8.00 and 9.30 a.m. to avoid diurnal variations. All mice were monitored intensely on the day of dosing, and any survivors were monitored for a further 14 d with bodyweight and food consumption being measured regularly. After 14 d, the mice were euthanized with carbon dioxide and necropsied. Major organs (liver, spleen, kidney, heart, lungs, gut) were immediately weighed and expressed as a percentage of bodyweight (Appendix A).

## Figures and Tables

**Figure 1 marinedrugs-23-00103-f001:**
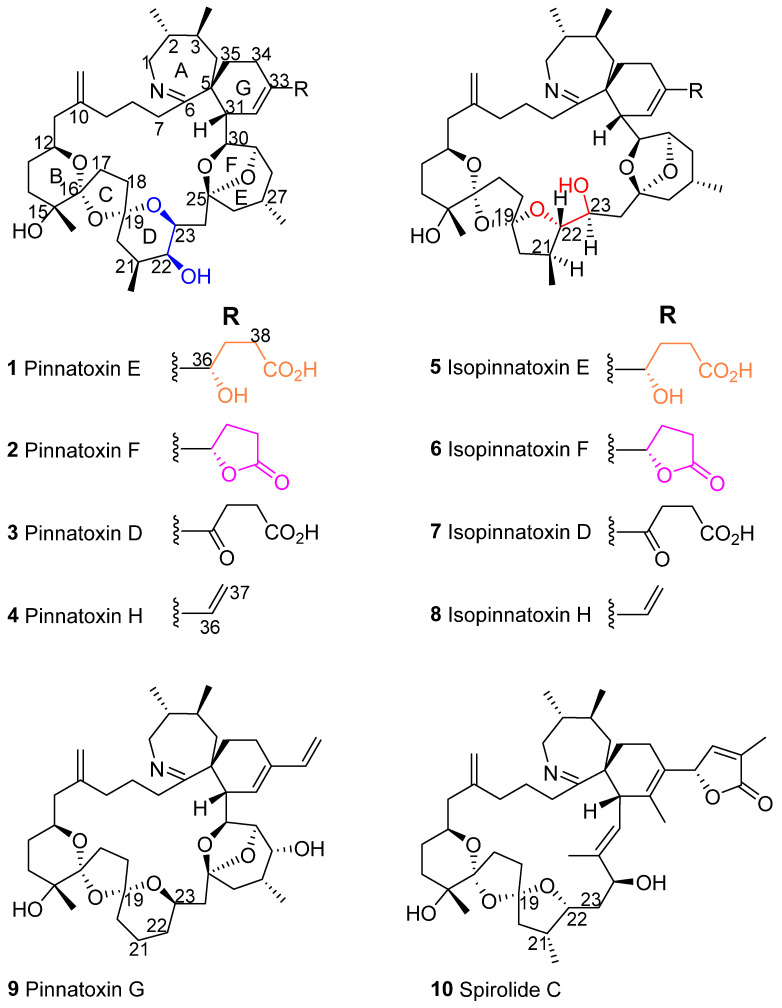
Chemical structures of pinnatoxins and spirolide C. The substructures colored blue and red, and orange and pink indicate the two interconverting substructures of these molecules (see Figure 4).

**Figure 2 marinedrugs-23-00103-f002:**
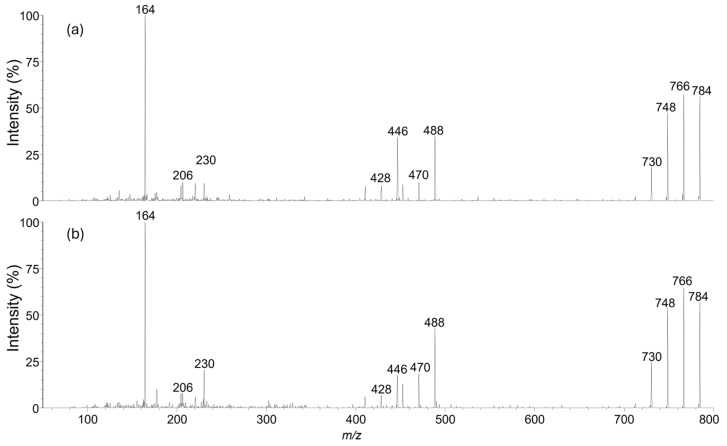
Collison-induced fragmentation spectra of **5** (**a**) and **1** (**b**).

**Figure 3 marinedrugs-23-00103-f003:**
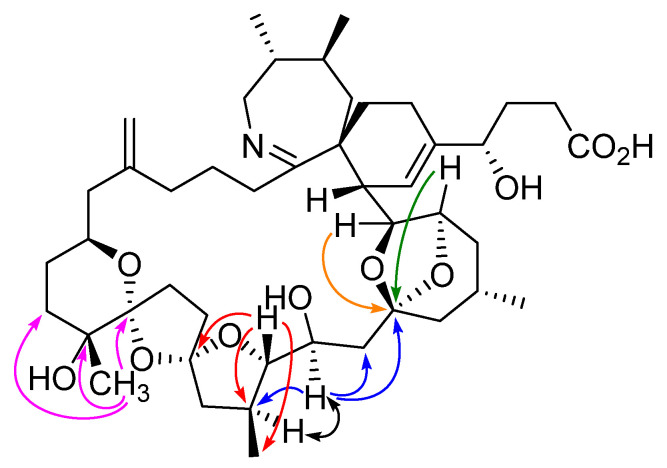
Structurally important HMBC (colored arrows) and NOESY (black arrow) correlations linking ether rings in compound **5**.

**Figure 4 marinedrugs-23-00103-f004:**
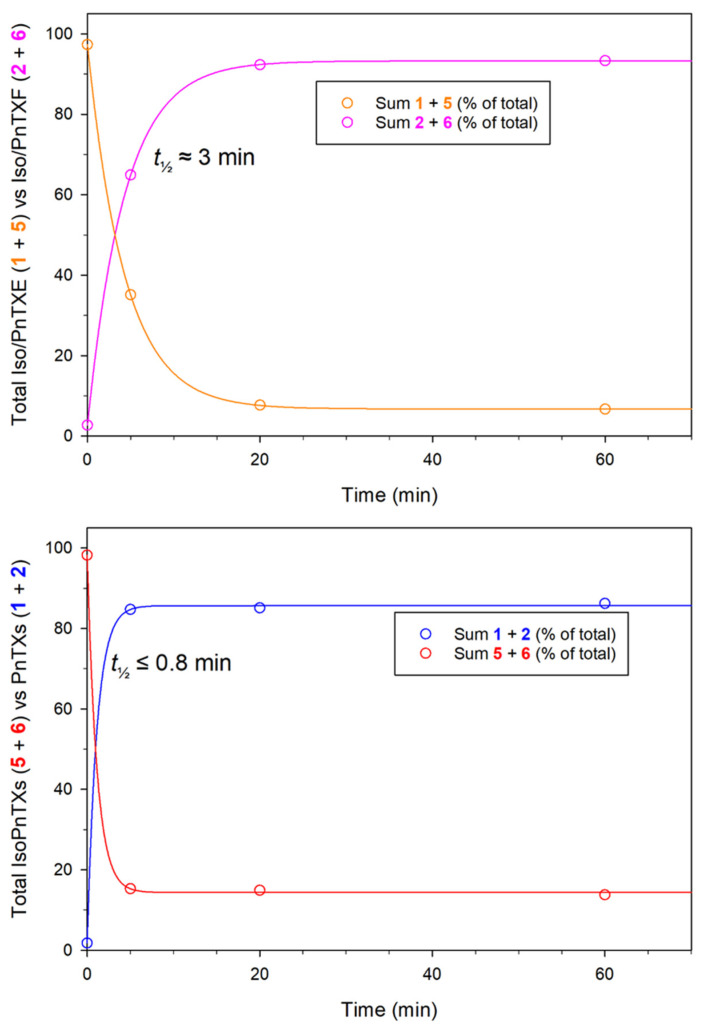
Mole fractions (%) of the four structural forms of pinnatoxin E (**1**) in the pinnatoxin (**1** and **2**) versus isopinnatoxin (**5** and **6**) forms, and in the lactone (**2** and **6**) versus γ-hydroxycarboxylic acid (**1** and **5**) forms, with time, obtained after the addition of isopinnatoxin E (**5**) to aqueous MeCN containing TFA. Data were fitted to three-parameter exponential curves to provide equilibrium positions (Table 2) and half-lives for the isomerization reactions. Similar experiments were also performed to evaluate the equilibration of **1**, **2**, **3**, **4**, and **9** (Table 2, Appendix A).

**Figure 5 marinedrugs-23-00103-f005:**
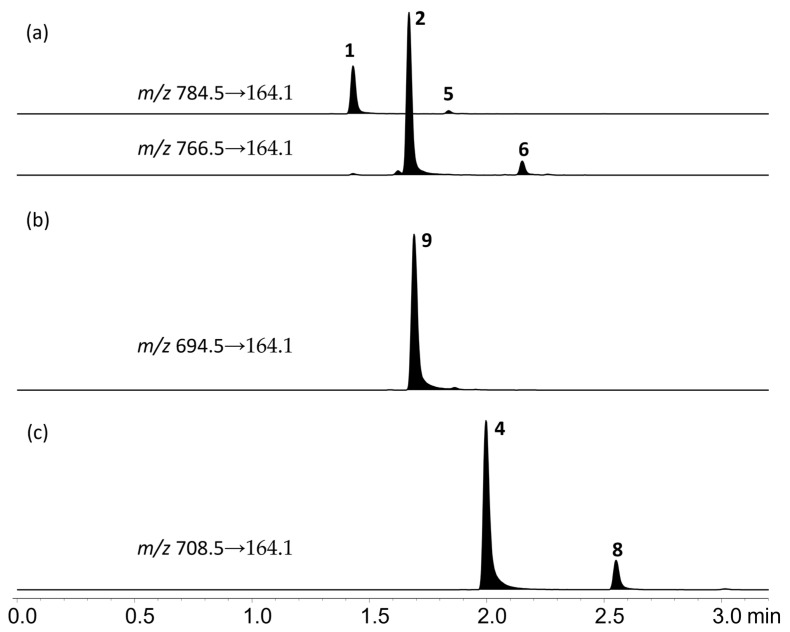
LC–MS/MS MRM chromatograms of extracts from three strains of *V. rugosum* showing pinnatoxin profiles: (**a**) strain CAWD167, containing pinnatoxins E (**1**), F (**2**), and their isomers (**5** and **6**); (**b**) CAWD188, contains pinnatoxin G (**9**), and; (**c**) CAWD198, containing pinnatoxin H (**4**) and its isomer (**8**). In addition to the spiroketal rearrangement, the γ-butyrolactone–γ-hydroxybutanoic acid moiety in compounds **1**, **2**, **5,** and **6** opened and closed to an equilibrated mixture of 94:6 lactone–hydroxyacid (Table 2, Figure 4 and Appendix A). Due to the occurrence of these acid-catalyzed rearrangements, compounds **1**, **2,** and **5** ended up as an equilibrated mixture of compounds **2**, **6**, **1**, and **5** at a ratio of approximately 80:14:5:1. The rate of the D-ring rearrangement was similar to or perhaps slightly faster than the ring closure of the hydroxy acid to the lactone under these conditions (Figure 4 and Appendix A).

**Table 1 marinedrugs-23-00103-t001:** NMR chemical shifts (ppm) of isopinnatoxin E (**5**) and pinnatoxin E (**1**) in CD_3_OD, and the C-1 to C-22 region of spirolide C (**10**) in CD_3_OH.

	5		1 ^a^		10 ^b^	
No.	^13^C	^1^H	^13^C	^1^H	^13^C	^1^H
1	53.7	3.49, 3.87	52.6	3.56, 4.10	53.3	3.44, 3.76
2	42.6	1.41	40.7	1.57	41.2	1.36
2-Me	20.3	1.14	20.3	1.18	19.4	0.98
3	34.3	1.63	35.5	1.30	36.9	1.16
3-Me	20.9	0.93	21.7	1.03	21.1	0.95
4	37.3	1.47, 1.74	36.5	1.66, 1.93	38.3	1.55, 1.73
5	51.1		51.9		50.8	
6	179.9		nd		178.6	
7	35.7	3.27	35.1	3.53	35.6	2.32, 2.43
8	23.1	1.43, 2.02	21.8	1.74, 2.15	23.4	1.40, 2.02
9	35.2	1.67, 2.35	33.8	1.82, 2.17	36.0	1.58, 2.10
10	148.3		146.5		147.8	
10=CH_2_	110.6	4.74, 4.76	111.3	4.81, 4.86	111.3	4.75, 4.78
11	47.0	2.05, 2.40	46.9	2.15, 2.37	47.6	2.01, 2.37
12	69.8	4.25	69.5	4.06	69.3	3.97
13	29.8	1.23, 1.65	29.8	1.27, 1.64	30.2	1.24, 1.55
14	35.3	1.48, 1.93	35.5	1.50, 1.90	35.8	1.49, 1.81
15	71.5		71.1		71.1	
15-Me	22.8	1.22	22.9	1.22	22.5	1.19
16	112.4		113.7		112.5	
17	32.5	1.81, 2.24	31.4	1.78, 2.20	31.5	1.74, 2.11
18	36.1	1.99, 2.17	38.7	1.85, 2.15	36.5	2.04, 2.22
19	116.8		110.1		117.4	
20	46.9	1.71, 2.40	37.7	1.66, 1.66	45.8	2.14, 2.26
21	35.7	2.51	32.1	2.09	35.4	2.41
21-Me	18.1	1.09	18.3	0.97	15.6	1.19
22	91.4	3.32	69.4	3.44	81.7	4.31
23	70.8	3.83 ^c^	73.1	4.09		
24	44.0	1.82, 1.82	40.1	1.92, 2.21		
25	110.1		109.6			
26	44.2	1.31, 2.21	45.4	1.41, 1.87		
27	25.8	2.35	26.0	2.29		
27-Me	22.8	0.98	22.7	0.99		
28	34.1	1.52, 2.01	33.8	1.55, 2.04		
29	77.7	4.57	77.4	4.62		
30	81.8	3.87	81.1	3.86		
31	43.5	3.36	44.7	3.52		
32	121.7	5.14	121	5.21		
33	144.2		144.3			
34	21.4	2.16, 2.16	21.9	2.26, 2.26		
35	33.7	1.68, 1.84	34.1	1.82, 1.92		
36	76.6	3.98	75.7	4.02		
37	32.8	1.82, 1.82	32.1	1.84, 1.84		
38	35.6	2.21, 2.21	33.7	2.25, 2.25		
39	182.3		180.3			

^a^ Assignments for compound **1** are from [2]. ^b^ Selected assignments for compound **10** are from [19]. ^c^ Doublet of doublets of doublets, *J* = 8.2, 6.0, 3.3 Hz.

**Table 2 marinedrugs-23-00103-t002:** Mole fractions (%) of various pinnatoxins in the pinnatoxin versus isopinnatoxin form, and in the lactone versus the γ-hydroxycarboxylic acid form, at equilibrium in aqueous MeCN containing TFA, obtained by fitting kinetic data for each pinnatoxin used (Figure 4 and Appendix A) to 3-parameter exponential curves *.

Pinnatoxin Used	Pinnatoxin	Isopinnatoxin	Carboxylic Acid	Lactone
Pinnatoxin E (**1**)	84.4	15.6	5.4	94.6
Pinnatoxin F (**2**)	84.8	15.2	5.4	94.6
Pinnatoxin D (**3**)	85.5	14.5	N/A	N/A
Pinnatoxin H (**4**)	85.9	14.1	N/A	N/A
Isopinnatoxin E (**5**)	85.7	14.3	6.8	93.2
Average for **1**–**5**	85.3	14.7	5.9	94.1
Pinnatoxin G (**9**)	100.0	N/A	N/A	N/A

* For equilibrating isomer pairs, estimated half-lives for attainment of equilibrium ranged from under ca. 0.8 to 3 min (Figure 4 and Appendix A), but reactions were too rapid to measure accurately. NA, not applicable, as these structures do not exist for these compounds.

## Data Availability

Data are contained within the article.

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
