# Peer review of "Structural Characterization of Pinnatoxin Isomers"

_marinedrugs, 2025, doi:10.3390/md23030103_

Round 1

Reviewer 1 Report

Comments and Suggestions for Authors

The manuscript is well structured and organized with relevant information.

We recommend improvements in small details that are attached in the PDF manuscript file, with some indexed suggestions.

There is scientific rigor in the information provided.

To improve the S9 figure, the blank letter text is not visible at the bottom of the page, please modify the color.

Table S1 standardizes the significant figures.  Apply the same magnitude for the values reported in the table.

Author Response

Comment 1: To improve the S9 figure, the blank letter text is not visible at the bottom of the page, please modify the color

Response 1: We agree and have changed the color of the font to white of the darker parts of the structure and black on lighter parts of the structure.

Comment 2: Table S1 standardizes the significant figures.  Apply the same magnitude for the values reported in the table.

Response 2: Thanks for pointing this out. We found that one value was not rounded correctly. We have corrected this and now all values are reported to two significant figures. 2.19% changed to 2.2%

Comment 3: Suggestion – change globe to world or worldwide “..s have been found in numerous iso- lates from around the globe..”

Response 3: We agree with the suggestion and have changed “globe” to world.

Comment 4: The toxicology of cyclic imines in humans is not at all clear, there is no evidence of human poisoning. Alos, these toxins are not regulated in the EU, there are no recommendations in the EFSA. The authors are recommended to describe in detail the importance of this study, either as a contribution to new knowledge in the field of toxicology or then organic chemistry/natural compounds.

Comment 4: The toxicology of cyclic imines in humans is not at all clear, there is no evidence of human poisoning. Alos, these toxins are not regulated in the EU, there are no recommendations in the EFSA. The authors are recommended to describe in detail the importance of this study, either as a contribution to new knowledge in the field of toxicology or then organic chemistry/natural compounds.

Response 4: We do indeed describe the importance of this study in terms of its contribution to new knowledge in the field of toxicology or then organic chemistry/natural compounds. In the article we describe the instability of 22-hydroxypinnatoxins, the reason for this instability, we identify the reaction products, and we measured the toxicological consequences of the isomerization reaction. We have also now included a paragraph regarding human guidance values: “Two different health-based guidance values have been proposed for pinnatoxins in shellfish, 23 μg/kg by Arnich et al. [22] and 268 μg/kg by Finch et al. [23] with the discrepancy due to the application of different uncertainty factors. By i.p injection 1, 2, 9 and 4 are 27, 98, 25 and 19 times more toxic than 5, respectively. Considering the low toxicity of isopinnatoxins, along with their low abundance, it is unlikely that they could make any meaningful contribution towards the health guidance figures set for total pinnatoxins.”

  1. Arnich, N.; Abadie, E.; Delcourt, N.; Fessard, V.; Fremy, J.M.; Hort, V.; Lagrange, E.; Maignien, T.; Molgó, J.; Peyrat, M.B.; et al. Health risk assessment related to pinnatoxins in French shellfish. Toxicon 2020, 180, 1-10, https://doi.org/10.1016/j.toxicon.2020.03.007
  2. Finch, S.C.; Harwood, D.T.; Boundy, M.J.; Selwood, A.I. A Review of Cyclic Imines in Shellfish: Worldwide Occurrence, Toxicity and Assessment of the Risk to Consumers. Mar Drugs 2024, 22, 129, https://doi.org/10.3390/md22030129.

Comment 5: The authors should reinforce the importance of this study. If it is a contribution to protecting public health consumers, are these toxins an imminent risk to public health?

For example, a series of Azaspiracid compounds were identified and characterized; however, its control is focused only on AZA1, AZA2 and AZA3.

Will this situation be identical with these new cyclic imine compounds?

Response 5: As scientists we cannot predict what regulatory bodies will decide, we present data that regulators can use to make informed decisions, but the outcomes sometimes appear illogical. Our conclusion in the discussion section stated the followed: “Considering the low toxicity of isopinnatoxins, along with their low abundance, it is unlikely that they could contribute significantly to the total toxicity of pinnatoxins.” We decided to move this to the abstract and have included the following paragraph to the discussion as mentioned in the response to your comment 4:

“Two different health-based guidance values have been proposed for pinnatoxins in shellfish, 23 μg/kg by Arnich et al. [22] and 268 μg/kg by Finch et al. [23] with the discrepancy due to the application of different uncertainty factors. By i.p injection 1, 2, 9 and 4 are 27, 98, 25 and 19 times more toxic than 5, respectively. Considering the low toxicity of isopinnatoxins, along with their low abundance, it is unlikely that they could make any meaningful contribution towards the health guidance figures set for total pinnatoxins.”

Based off the data we have provided in this manuscript if pinnatoxins were to be regulated, it would be easy to incorporate a relative toxicity factor to account for the inevitable presence of isopinnatoxins.

Comment 6: Suggestion - With the experimental data from this study, authors could indicate equivalent toxicity factor for this isomers? Such as PST analogues are expressed as STX equivalent. Could this type of information be provided with tthe results of your experiments?

Comment 6: Suggestion - With the experimental data from this study, authors could indicate equivalent toxicity factor for this isomers? Such as PST analogues are expressed as STX equivalent. Could this type of information be provided with tthe results of your experiments?

Response 6: We have provided an LD50 value for isopinnatoxin E, so it pretty straight forward to calculate a TEF to another pinnatoxin if pinnatoxins were to become regulated. We added information about the difference in toxicity of isopinnatoxin E relative to other pinnatoxins in the new paragraph that we mentioned in the response to your comment 4. 

Comment 7: To complete this study, shellfish should be feed with this species of dinoflagellates used in this study and then evaluate the bioaccumulation and bioconversion of these pinatoxins. Why was this study not carried out to complete the study of isolation, characterization of the new isomers of pinatoxin?

Response 7: Thanks for the question. A shellfish feeding study is beyond the scope of this study as outlined in the introduction, which was stated to be to identify the unknown pinnatoxin isomers previously detected in a range of natural samples (including but not limited to shellfish).  We believe that such an experiment would provide no relevant additional information about isopinnatoxins. The work we reported here clearly shows that the isopinnatoxins are not generated by metabolism in shellfish but, instead are an inevitable result when a 22-hydroxy group is present in pinnatoxins.

Comment 8: This final conclusion is very important and relevant. Authors should incorporate and highlight this observation in the abstract of the manuscript.  “Considering the low toxicity of isopinnatoxins, along with their low abundance, it is unlikely that they could contribute significantly to the total toxicity of pinnatoxins.”

Response 8: Thanks for pointing out that this was not clearly stated in the abstract. We initially thought the conclusion was evident in the last sentence, but we are happy to modify the text as follows to make it clearer. As mentioned in our response to your comment 5, we have moved the following sentence from the discussion to the abstract: Considering the low toxicity of isopinnatoxins, along with their low abundance, it is unlikely that they could contribute significantly to the total toxicity of pinnatoxins.”

Comment 9: Recommendation:

For reagents nomenclature always use the full name or an abbreviated formula, try to use uniform criteria in all manuscripts.

It is recommended, for example, first use the full name and then the short, Methanol (MeOH), ... acetonitrile (MeCN)... etc.

Proportions of masses, volumes, etc. should be indicated in subscript, e.g. 0.1%(v/v) acetic acid... it is recommended to follow the IUPAC criteria for expressing mass, volumetric quantities in solutions and mixtures.

Response 9: MeOH and MeCN are not generally considered abbreviations and in chemistry journals typically do not need to be identified as such. They are representations of chemicals structures (methyl alcohol and methyl cyanide). However, it is not clear in the instructions to authors if these are considered to be abbreviations or not. So, we can include the common names in the first instance (methanol and acetonitrile). 

We have added v/v to %concentrations and/or changed to volume ratios where volume plus volume were used, as is common with chromatography.

Comment 10: acetic acid solution? which concentration? or was pure reagent? “..pH was adjusted to pH 5.5 using acetic acid.”

Response 10: Acetic acid = pure acetic acid, if it was a dilute solution then the concentration would have been stated.

Comment 11: Explain very briefly why the qualitative transition 164.1 has been chosen for monitoring in mass acquisition. It is useful information for readers.

Response 11: We have added the following sentence: ‘The m/z 164.1 ion corresponds to the common iminium ring fragment (C11H18N+).

Comment 12: Line 227. "on the day" is ambiguous.  24 h??

‘All mice were monitored intensely on the day of dosing and any survivors were monitored for a further 14 d with bodyweight and food consumption being measured regularly.’

Response 12: We agree and have changed the sentence to the following: “All mice were monitored continuously for the first hour post-dosing and then every 1-2 hours until 8 h post-dosing.  Any survivors were weighed every 1-2 d and food consumption measured”.

Reviewer 2 Report

Comments and Suggestions for Authors

This is a well-designed study with very interesting findings on determining the chemical structures of isomers of pinnatoxins D,E,F and G, namely isopinnatoxins, by means of a combination of LC–MS/MS and NMR spectroscopy and studying the acute toxicity of isopinnatoxin E. There are only few points needing some improvement in order to make the manuscript more reader friendly.

General remarks:

The only weak point in the manuscript is the Discussion section, which needs to be further enriched according to the comments provided in the relevant section. Other minor points are detailed below:

Specific remarks:

  1. Results

- Page 4, lines 72-73 – Table 1: Please indicate in the table caption that spicolide C corresponds to compound “10”. It is not clear.

- Page 7, lines 112-113 – Figure 5: Please indicate in the 766.5 →164.1 chromatogram what compound corresponds to the highest peak – according to RT it probably is 9, but it is not certain. Also, please provide the compounds explanation in the figure caption, so that the figure can stand alone without referring to the text.

- Page 7, line 116: Please correct “occurance” to “occurrence”.

  1. Discussion

- The discussion section is very limited and needs to be enriched. It is suggested to discuss the similarities of the isopinnatoxin E to spirolide C (derived from the NMR spectrum) and potential interpretation of this finding (in terms of origin, chemistry, whatever may be relevant).

- Page 8, lines 154-155: It is suggested to discuss the toxicity of IsoPnTX E and its potential contribution taking into account the provisional acute health-based guidance value for PnTX G of 23 μg/kg proposed by ANSES.  

  1. Materials and Methods

- Page 8, line 180: “Fractions 1012”: something is wrong here, please correct.

- Page 8, lines 183-184: Please provide the particle size for the preparative column.

- Page 9, line 229: Please clarify what “regularly” means in this case. What was the frequency of measurement?

Supporting information

- Page S17, Table S1: Please indicate at what point were the weights measured? I assume it is the day that the mice were sacrificed but it should be indicated in the table caption.

Author Response

Comment 1: Page 4, lines 72-73 – Table 1: Please indicate in the table caption that spicolide C corresponds to compound “10”. It is not clear.

Response 1: We agree that it was not clear. We have included the compound number (10) after the compound name, spirolide C.

Comment 2: Page 7, lines 112-113 – Figure 5: Please indicate in the 766.5 →164.1 chromatogram what compound corresponds to the highest peak – according to RT it probably is 9, but it is not certain. Also, please provide the compounds explanation in the figure caption, so that the figure can stand alone without referring to the text.

Response 2: Thanks for pointing that out. The compound number was cut out when converting to a picture format. We have included the compound number to show that the largest peak is pinnatoxin F (2) and have included additional text in the caption to summarize which pinnatoxins were found in the various strain extracts: “Figure 5. LC–MS/MS MRM chromatograms of extracts from three strains of V. rugosum showing pinnatoxin profiles: (a) Strain CAWD167, containing pinnatoxins E (1), F (2) and their isomers (5 and 6); (b) CAWD188, contains pinnatoxin G (9), and; (c) CAWD198, containing pinnatoxin H (4) and its isomer (8).

Comment 3: Page 7, line 116: Please correct “occurance” to “occurrence”.

Response 3: Thanks. We have corrected this spelling mistake.

Comment 4: The discussion section is very limited and needs to be enriched. It is suggested to discuss the similarities of the isopinnatoxin E to spirolide C (derived from the NMR spectrum) and potential interpretation of this finding (in terms of origin, chemistry, whatever may be relevant).

Response 4: Thanks for the suggestion, but there is no potential interpretation to be made here in terms of the origin, chemistry of anything else we can think of. The spirolide C NMR chemical shifts were included as reference chemical shifts for the configuration of the 6-5-5 spiroketal ring system and we state in the NMR results that where changes in the chemical shifts for isopinnatoxin E differed from pinnatoxin E, they more closely resembled spirolide C. This aligns with the assigned structure of a 6-5-5 spiro ring system.  

Comment 5: Page 8, lines 154-155: It is suggested to discuss the toxicity of IsoPnTX E and its potential contribution taking into account the provisional acute health-based guidance value for PnTX G of 23 μg/kg proposed by ANSES.  

Response 5: Thanks for the suggestion we have included the following text: “Two different health-based guidance values have been proposed for pinnatoxins in shellfish, 23 μg/kg by Arnich et al. [1] and 268 μg/kg by Finch et al. [2] with the discrepancy due to the application of different uncertainty factors. By i.p injection 1, 2, 4 and 9 are 22, 98, 19 and 26 times more toxic than 5, respectively.” Considering the low toxicity of isopinnatoxins, along with their low abundance, it is unlikely that they could make any meaningful contribution towards the health guidance figures set for total pinnatoxins.”

Comment 6: Page 8, line 180: “Fractions 1012”: something is wrong here, please correct.

Response 6: This typo has been corrected. “Fractions 1012  were combined..”

Comment 7: Page 8, lines 183-184: Please provide the particle size for the preparative column.

Response 7: We have added the particle size (10 µm) of the column packing to this text.

Comment 8: Page 9, line 229: Please clarify what “regularly” means in this case. What was the frequency of measurement?

Response 8: We agree and have changed the sentence to the following: “All mice were monitored continuously for the first hour post-dosing and then every 1-2 hours until 8 h post-dosing.  Any survivors were weighed every 1-2 d and food consumption measured”.

Comment 9: Page S17, Table S1: Please indicate at what point were the weights measured? I assume it is the day that the mice were sacrificed but it should be indicated in the table caption.

Response 9: The organs were weighed immediately after necropsy. We have changed the text in the methods and materials section to include the word immediately: “After 14 d the mice were euthanized with carbon dioxide and necropsied. Major organs (liver, spleen, kidney, heart, lungs, gut) were immediately weighed and expressed as a percentage of bodyweight (Table S1).”

Reviewer 3 Report

Comments and Suggestions for Authors

In my opinion, the paper is interesting, correct and well written. I can only suggest two minor changes. 

Line 212. Explain how much the aliquots were diluted and the solvent used.

Line 227. "on the day" is ambiguous.  24 h??

Author Response

Comments 1: Line 212. Explain how much the aliquots were diluted and the solvent used.

Response 1: We have re-written this sentence to improve clarity around how the dilutions were prepared: “These samples were diluted 100-fold by adding 10 µL aliquots to 990 µL of MeCN–water (1:1 v/v) and analyzing immediately by LC-MS/MS.”

Comment 2: Line 227. "on the day" is ambiguous.  24 h??

Response 2: Thanks for pointing this out. We have updated the text to: “All mice were monitored continuously for the first hour post-dosing and then every 1-2 hours until 8 h post-dosing.  Any survivors were weighed every 1-2 d and food consumption measured”